# Predicting depression among men who have sex with men in Ghana using machine learning algorithms

Abdulzeid Yen Anafo[1]*, LaRon E. Nelson[2,3], Leo Wilton[4,5], Vincent Uwumboriyhie Gmayinaam[6], Selasi Ocloo[7]

1 Department of Mathematical Sciences, University of Mines and Technology, Tarkwa, Ghana, 2 School of Nursing, Yale University, New Haven, Connecticut, United States of America, 3 Rotman School of Management, University of Toronto, Toronto, Ontatio, Canada, 4 Department of Human Development, State University of New York at Binghamton, Binghamton, New York, United States of America, 5 Faculty of Humanities, University of Johannesburg, Johannesburg, South Africa, 6 Department of Biostatics and Epidemiology, University of Health and Allied Science, Hohoe, Ghana, 7 Department of Engineering, Ashesi University, Brekuso, Ghana

* ayanafo@umat.edu.gh

## Abstract

Men who have sex with men (MSM) in Ghana face heightened risks of depression due to pervasive stigma, social exclusion, and legal discrimination. Despite this, depression remains underdiagnosed and undertreated in this population. This study applied seven tree-based machine learning (ML) models using tree-based classifiers: Decision Tree, Random Forest, Gradient Boosting, AdaBoost, XGBoost, LightGBM, and CatBoost to identify key psychosocial predictors of depression in a sample of 225 MSM aged 18–60 years. The dataset included sociodemographic variables, perceived stress (PSS), social isolation (internal and external), behavioural risk indicators, and stigma-related measures. After handling missing values, data were pre-processed with feature standardization and one-hot encoding. The Synthetic Minority Over-Sampling Technique was applied to address class imbalance. Model performance was evaluated using 5-fold cross-validation and metrics such as accuracy, precision, recall, F1 score, and ROC AUC. Among all models, Random Forest achieved the highest accuracy for the prediction of depression amongst MSM in Ghana. Feature importance analysis revealed that external social isolation (ExtSocialIso2), perceived stress (PSS14), and stigma due to same-sex behaviour (StigmaSSB9) were the most consistent predictors of depression. Variables related to resilience, gender non-conformity stigma, and sense of community belonging also contributed significantly. Depression among MSM in Ghana is closely linked to social isolation, stress, and identity-based stigma. Machine learning models, especially ensemble methods, can effectively identify individuals at risk. These findings underscore the need for culturally tailored mental health interventions and inclusive policies that address stigma and promote social support among MSM in Ghana.

**Data availability statement:** The datasets used and/or analysed during the current study are available from https://doi.org/10.5064/F6QKAMIP.

**Funding:** The authors received no specific funding for this work.

**Competing interests:** The authors have declared that no competing interests exist.

## Introduction

Multiple studies and surveys have consistently shown higher rates of depression among men who have sex with men (MSM) compared to their heterosexual counterparts [1–4]. The World Health Organization (WHO) estimates that 3.8% of the global population experiences depression, with higher rates among women (5.1%) compared to men (3.6%) in the general population [5]. The rates among gay men significantly exceed these global averages. Studies indicate that mental health disparities exist among sexual minorities globally, including in Ghana and West Africa. The World Health Organization (WHO) highlights that sexual minority groups often face increased mental health risks due to stigma, discrimination, and social exclusion, contributing significantly to higher rates of depression and anxiety [6]. In Ghana specifically, studies have demonstrated elevated mental health challenges among MSM. For instance, related studies on stigma and depression by [7–9] found that MSM in Ghana exhibited notably higher rates of depressive symptoms and psychological distress compared to the general population. Despite these findings, there remains limited research specifically addressing mental health issues such as loneliness and depression within MSM populations across West Africa. Further investigation is crucial to understanding the local context and developing targeted public health interventions to support MSM communities effectively.

Machine learning is a branch of artificial intelligence that focuses on making programs using datasets to find patterns, make decisions, and guess what will happen without being explicitly programmed to do each job [10]. Machine learning algorithms typically "learn" by examining historical data, which enables them to continuously improve their accuracy and effectiveness as more data become available [11]. Studies over the past decades have widely explored the use of machine learning algorithms to predict and detect various mental health conditions, including depression and anxiety. Many of these studies have focused on leveraging social media data and other digital footprints to identify markers of mental health issues. For instance, studies have utilised techniques like topic modelling, bag-of-words, and tf-idf to analyze text from blog posts and categorize symptoms of anxiety and depression [12,13]. Convolutional neural networks (CNNs) have been found to outperform other classifiers in accurately predicting depression and suicide risk from social media data [14]. Audio-based approaches, such as using long short-term memory (LSTM) neural networks on speech samples, have also shown promise in detecting depression [15]. While the application of machine learning in mental health prediction has been more extensively studied in Western contexts, a growing body of research is also exploring its potential in African and sub-Saharan African settings. A study conducted in Nigeria utilized logistic regression, Naïve Bayes, and random forest classifiers to predict anxiety and depression among seafarers, using demographic and health-related factors as input features [16]. The random forest model achieved the highest accuracy for the two datasets analyzed. Another study from South Africa applied supervised machine learning techniques to detect post-traumatic stress disorder (PTSD) among ex-servicemen, using alcohol misuse, gender, and deployment status as predictors [17].

The performance of different machine learning algorithms has been found to vary depending on the specific scenario and context. No single algorithm has been determined as the most suitable in all cases [14–16]. Given the limited research on machine learning to predict depression and loneliness among MSM in Ghana, there is a clear need for further exploration in this area. In the present study, we applied a range of machine-learning algorithms to identify the symptoms of depression and loneliness in this population. The main contributions of this study will include exploring the application of machine learning algorithms to predict depression among MSM in Ghana accurately and addressing the limitations of traditional diagnostic methods. The aim of this study is to conduct a comparative assessment of various machine learning models, including tree-based classifiers: Decision Tree, Random Forest, Gradient Boosting, AdaBoost, XGBoost, LightGBM, and CatBoost, to ascertain which algorithms exhibit superior performance in predicting mental health disorders based on established metrics. The study also seeks to discover key predictive traits, including emotional isolation, anhedonia, and low mood, by analysing several mental health indicators, thereby offering greater insights into the variables impacting vulnerability among MSM in Ghana. While traditional statistical methods are effective for identifying associations between psychosocial variables and mental health outcomes, they often rely on strict assumptions about linearity, normality, and multicollinearity. These limitations can hinder the detection of complex, non-linear interactions among multiple risk factors interactions that are likely present in multifaceted issues such as depression among MSM populations. ML, by contrast, offers flexible, data-driven techniques capable of uncovering hidden patterns in high-dimensional data without pre-specifying relationships. This makes ML particularly well-suited for developing predictive models that can generalize to unseen individuals and settings. In resource-limited contexts such as Ghana, ML-based screening tools could serve as scalable, low-cost alternatives to clinical evaluations, supporting early detection, triaging, and targeted interventions. The predictive focus of this study, therefore, not only aims to enhance understanding of depression risk among MSM but also to lay the groundwork for future deployment of digital health tools that can identify at-risk individuals in real time, even outside of traditional healthcare environments. This is critical in settings where stigma and discrimination hinder access to mental health services, and where data-driven, anonymous tools could support community outreach and policy planning.

## Methodology

### Design

This research focused on detecting depression using the Patient Health Questionnaire (PHQ) 1 and 2 scores from secondary data collected in a cross-sectional bio-behavioral study. The purpose of the parent study was to investigate association between stigmas and the various components of the HIV care continuum, including diagnoses, linkage to care, retention, and viral load suppression [18]. The parent study was conducted between 2016-2017. Data were collected from a total of 225 participants and subsequently classified using seven machine learning algorithms: Random Forest Tree, Decision Tree, Gradient Boosting, AdaBoost, XGBoost, LightGBM and CatBoost.

### Ethics statement

The University of Rochester's Research Subjects Review Board granted approval (Approval Number: 60120), alongside the Kwame Nkrumah University of Science & Technology's Committee on Human Research, Publication, and Ethics in Kumasi, Ghana (approval Number CHRPE/AP/523/16). Participants were prospectively recruited between 30 December 2016 and 4 May 2017. All participants submitted written informed consent before joining the study. Research assistants collected electronic written consent instead of paper forms, storing them in REDCap to ensure subjects' privacy. The consent forms were available in English, which is Ghana's primary written language.

### Participants

This study included 225 men who have sex with men (MSM), aged 18 to 60 years, residing in Ghana [18]. Participants represented a diverse group, including both employed and unemployed individuals with responsibilities ranging from

household duties to professional roles. The study was conducted in four major cities: Accra, Kumasi, Takoradi, and Koforidua. Eligible participants were cisgender men who had engaged in sexual contact with another man at least once in their lifetime and had been diagnosed with HIV for at least six months.

**Measures for predictive modelling**

**Inputs variables.** Predictor variables for the classification of depression were informed by data availability and insights from previous research [19–22]. S1 Table illustrates variables encompassing sociodemographic and psychosocial characteristics, including perceived stress levels, social isolation metrics, behavioural risks, stigma associated with same-sex behaviour, stigma related to gender non-conformity, marital status, education, religion, sexual attraction, and categorised age and income groups. Sociodemographic characteristics encompass educational attainment (D4EducLevel), married status (D8Married), religious affiliation (D9Religion), and sexual orientation (D7Sex_attract). Income and age were categorised into bins: Low, Middle, and High for income; Young, Middle-aged, and Older for age to more effectively identify potential nonlinear relationships with depression risk.

Psychosocial factors included scores from the 14-item Perceived Stress Scale (PSS1 to PSS14) and assessments of both external and internal social isolation (ExtSocialIso1 to ExtSocialIso8, IntSocialIso1 to IntSocialIso13), Behavioural Risk (BRScale1 to BRScale6), Same-Sex Behaviour Stigma (StigmaSSB1 to StigmaSSB10), and Gender Non-Conformity Stigma (StigmaGNC1 to StigmaGNC13). These attributes provide an exhaustive explanation of the individual's psychological and social health (S1 Table).

**Output variable for predictive modelling.** The Patient Health Questionnaire-2 (PHQ-2) was modelled as the output variable, brief as a screening tool designed to assess the presence of depressive symptoms over the past two weeks. It consists of two items, each rated on a 4-point Likert scale (0 = Not at all, 1 = Several days, 2 = More than half the days, 3 = Nearly every day).

The two items of the PHQ-2 are:

1. Over the past two weeks, how often have you been bothered by little interest or pleasure in doing things? (PHQ1)

2. Over the past two weeks, how often have you been bothered by feeling down, depressed, or hopeless? (PHQ2)

The total score is obtained by summing the points for the two items, resulting in a score ranging from 0 to 6. Higher scores indicate a higher likelihood of depression, with a common cutoff of 3 or more points suggesting the need for further evaluation. The PHQ-2 serves as a quick initial screening for depression and is often followed by the more comprehensive PHQ-9 if the score indicates potential depressive symptoms.

**Data analysis**

The study analyses were performed using Python (version 3.11) alongside programs such as pandas [23], scikit-learn [24], imbalanced-learn [25], XGBoost [26], LightGBM [27], and CatBoost [28]. Missing numerical data has been replaced with the median. Categorical variables underwent one-hot encoding, whereas continuous variables were standardised utilising StandardScaler. The full dataset was initially partitioned into training (80%) and testing (20%) sets using stratified sampling to maintain the distribution of the target variable. All data preprocessing steps (imputation, SMOTE, encoding, and scaling) were applied only to the training set to prevent data leakage. The training set was then used to develop the models and tune hyperparameters using stratified 5-fold cross-validation. Model performance was evaluated on the test set, which remained untouched throughout the training process. Both cross-validation metrics (mean accuracy, F1, precision, and recall) and final test set performance metrics are reported. All models were trained using their default hyperparameters as implemented in Scikit-learn, XGBoost, LightGBM, and CatBoost libraries. This approach ensured a fair comparison of baseline performance across classifiers and minimized the risk of overfitting given the limited sample size.

Hyperparameter tuning, such as grid or randomised search, was intentionally not conducted due to the small sample size, to prevent model overfitting and inflated performance estimates, the study used pre-specified (untuned) settings for each model to provide fair, reproducible baselines. This decision aligns with best practices in predictive modeling when working with limited datasets and avoids data leakage from excessive model optimization. Feature importance was assessed using Gini importance, also called Mean Decrease in Impurity. This metric measures the overall role of each feature in reducing impurity across decision nodes. It is a default feature in all tree-based algorithms employed in this study.

**Feature selection.** Feature selection in this study was guided by a theory-driven rather than data-driven approach. Anchored in minority stress theory and prior empirical studies on depression among sexual minority populations, we selected variables based on their conceptual relevance to mental health disparities among men who have sex with men (MSM). Specifically, features were drawn from: Sociodemographic variables such as education level, marital status, religion, and categorized age and income groups. These were included to capture structural and social determinants of mental health. Psychosocial factors: The Perceived Stress Scale (PSS1–PSS14), a validated 14-item instrument, was included to assess subjective stress levels. External and internal social isolation items (ExtSocialIso1–8, IntSocialIso1–13) were selected based on their predictive role in mental distress among LGBTQ+ individuals. Items measuring stigma related to same-sex behavior (StigmaSSB1–10) and gender non-conformity (StigmaGNC1–13) were included in line with studies showing that identity-related stigma is a key contributor to depression. Behavioral risk indicators (BRScale1–6) and self-compassion (SCS1–8) were also incorporated to capture internal coping mechanisms and resilience factors (see S1 Table).

All selected features were retained without automated algorithmic feature reduction to preserve theoretically significant predictors, even if some demonstrated weak individual predictive power. Categorical variables were one-hot encoded, and all continuous variables were standardised before model fitting. This theory-based feature selection approach ensured that the machine learning models remained interpretable, domain-aligned, and contextually grounded.

## Data pre-processing and partitioning

During the pre-processing stage of the study, the median was used to fill in missing values in continuous variables because it can handle outliers. An independent, stratified 20% test was set to preserve the target distribution. All model development, preprocessing, and resampling were performed exclusively on the remaining 80% of the training data using stratified 5-fold cross-validation (CV). For each CV split, transformers were fitted on the training fold only, including numeric imputation (using the median), categorical encoding, and feature scaling. These models were then applied to the corresponding validation fold using the parameters learned from the training fold. The same transformers, fit on the full 80% training set, were later applied to the untouched 20% test set.

**Feature engineering.** Age and income, originally recorded as continuous variables, were grouped into categorical bins to reflect meaningful strata. Age was categorized into three groups: Young (≤25 years), Middle-aged (26–40 years), and Older (>40 years). Monthly income was similarly binned into Low (≤999 GHS), Middle (1,000–4,999 GHS), and High (≥5,000 GHS) categories. These variables were one-hot encoded, excluding the first category, to avoid multicollinearity.

**Categorical encoding and scaling.** One-hot encoding was applied to categorical variables such as income and age groups. Label encoding was used for the target variable, Depression_Status. All continuous features were standardized using z-score normalization via StandardScaler from scikit-learn to ensure uniform scaling and enhance model convergence, especially for models sensitive to feature magnitude.

**Handling class imbalance.** As the target variable was imbalanced, the training dataset was resampled using the Synthetic Minority Over-sampling Technique (SMOTE) to synthetically generate new examples in the minority class. Given the class imbalance (19.8% depressed vs. 80.2% not depressed), we applied SMOTE only within the training portion of each cross-validation split to balance the classes during model fitting; no resampling was performed on validation folds or on the held-out test set.

**Metric analysis.** Performance metrics used to evaluate the machine learning models included accuracy, precision, recall (also known as sensitivity), specificity, and the F1 score. These metrics were derived from confusion matrices generated for each model. Accuracy measures the overall correctness of predictions [29]. Precision assessed the proportion of true positive predictions among all predicted positives, and recall evaluated how effectively the models identified actual positive cases [30]. Specificity indicated the model's ability to accurately identify negative cases, whereas the F1 score provided a balanced measure combining precision and recall, offering a comprehensive evaluation of model performance [31]. These metrics were computed using true positive (TP), false negative (FN), false positive (FP), and true negative (TN) values extracted from the confusion matrices, providing detailed insights into each model's predictive capabilities [30]. The classification threshold for all models was set at the default value of 0.5, as implemented in the Scikit-learn and gradient boosting libraries. This decision ensured consistency across classifiers and enabled fair baseline comparisons. Although threshold tuning can improve individual metrics like sensitivity or specificity, we refrained from optimizing thresholds due to the small sample size and the exploratory nature of this study. Complementary metrics including precision, recall, F1-score, and ROC-AUC were reported to provide a more comprehensive evaluation of model performance, particularly considering the class imbalance present in the dataset

## Machine learning models

Seven machine learning algorithms were implemented using Python to predict the percentage of individuals experiencing depression symptoms. The Decision Tree algorithm models decision processes using a tree-like structure. It is intuitive and interpretable, making it a useful tool for psychological assessment and screening. Recent studies have applied decision trees to predict psychological disorders using questionnaire-based data. In psychological research, decision trees have been applied to analyze psychodiagnostics test data, offering informative, accurate, and efficient assessments [32]. Random Forest is an ensemble method that builds multiple decision trees and aggregates their outputs to improve generalization and reduce overfitting. It has been widely used in psychology for classifying disorders such as depression using behavioural and neuroimaging data [33]. Gradient Boosting builds models sequentially, where each new model attempts to correct the errors made by the previous ones. It has been applied to facial expression analysis in psychotherapy and clinical psychology research [34,35]. AdaBoost combines multiple weak learners into a strong classifier by focusing more on previously misclassified instances. It is less frequently used in psychology, but ensemble learning including AdaBoost is increasingly applied for psychological diagnostics and mental health prediction [36]. XGBoost is a powerful, regularized version of gradient boosting that is known for its speed and predictive accuracy. It has been applied in psychology for classifying individuals with major depressive disorder using brain imaging data [37]. LightGBM is optimized for speed and memory efficiency and is especially useful for large datasets. It has been used in psychology to model behavioral responses and predict anxiety or depression severity from survey responses [38,39]. CatBoost is a gradient boosting algorithm developed by Yandex that is particularly efficient at handling categorical variables. It utilises ordered boosting and symmetric tree structures to enhance training stability and mitigate overfitting. CatBoost handles categorical features natively and has shown promise in predicting mental health outcomes during complex events such as the COVID-19 lockdown [40].

## Reduced-feature sensitivity analysis

To assess whether the three most influential domains are sufficient on their own, we fit two classifiers, Logistic Regression (interpretable baseline) and Random Forest (top performer), using only perceived stress (PSS_StressLevel), external social isolation (ExtSocialIso_Level), and internal social isolation/trust (IntSocialIso_Level). The study employed the same pipeline as the main analysis: an independent stratified 20% test split. Within the remaining 80% of the training data, a stratified 5-fold cross-validation was applied. One-hot encoding and SMOTE were applied only to the training folds, while the test set remained untouched until the final evaluation.

## Results

### Classification report on training data

Table 1 presents the evaluation results on training data of seven classification models; Decision Tree, Random Forest, Gradient Boosting, AdaBoost, XGBoost, LightGBM, and CatBoost, assessed using both 5-fold cross-validation and confusion matrix-based performance metrics. A detailed graphical representation of the confusion matrices is shown in S1 Fig, while the complete 5-fold cross-validation results are provided in S2 Table. In this binary classification study, where the target labels are depressed (label 1) and not depressed (label 0) individuals, the Random Forest classifier emerged as the most effective model. It achieved the highest test accuracy of 80.3%, with a precision of 0.81, recall of 0.80, and F1-score of 0.80. It also recorded the highest mean cross-validation accuracy of 80.4%, highlighting its strong generalization ability and consistent performance across multiple folds. The confusion matrix for Random Forest revealed only two false negatives and six false positives, demonstrating its effectiveness in correctly identifying individuals who are depressed a critical requirement in mental health screening, where minimizing false negatives is essential.

CatBoost and Gradient Boosting also demonstrated strong performance. CatBoost achieved a test accuracy of 77.4%, with a precision of 0.77, recall of 0.77, and F1-score of 0.77. Gradient Boosting showed similar results, achieving a test

**Table 1. The table below summarizes the performance of each model using cross-validation on the balanced training dataset.**

| | Model | | | Accuracy | Precision | Recall | F1-Score | Mean Cross-Validation Accuracy |
|---|---|---|---|---|---|---|---|---|
| | Decision Tree | | | 69.66% | 0.6966 | 0.6966 | 0.6966 | 0.6963 |
| True label | | 0 | 1 | | | | | |
| | 0 | 9 | 5 | | 0.6983 | 0.6923 | 0.6953 | |
| | 1 | 10 | 21 | | 0.6949 | 0.7009 | 0.6979 | |
| | Random Forest | | | 80.34% | 0.8126 | 0.8034 | 0.802 | 0.8036 |
| True label | | 0 | 1 | | | | | |
| | 0 | 8 | 6 | | 0.866 | 0.7179 | 0.785 | |
| | 1 | 2 | 29 | | 0.7591 | 0.8889 | 0.8189 | |
| | Gradient Boosting | | | 76.07% | 0.767 | 0.7607 | 0.7593 | 0.7607 |
| True Label | | 0 | 1 | | | | | |
| | 0 | 10 | 4 | | 0.8081 | 0.6838 | 0.7407 | |
| | 1 | 6 | 25 | | 0.7259 | 0.8376 | 0.7778 | |
| | AdaBoost | | | 67.09% | 0.6731 | 0.6709 | 0.6699 | 0.6708 |
| True Label | | 0 | 1 | | | | | |
| | 0 | 10 | 4 | | 0.6923 | 0.6154 | 0.6516 | |
| | 1 | 9 | 22 | | 0.6538 | 0.7265 | 0.6883 | |
| | XGBoost | | | 76.07% | 0.7626 | 0.7607 | 0.7602 | 0.7609 |
| True Label | | 0 | 1 | | | | | |
| | 0 | 10 | 4 | | 0.785 | 0.7179 | 0.75 | |
| | 1 | 9 | 22 | | 0.7402 | 0.8034 | 0.7705 | |
| | LightGBM | | | 74.79% | 0.7493 | 0.7479 | 0.7475 | 0.7477 |
| True Label | | 0 | 1 | | | | | |
| | 0 | 10 | 4 | | 0.7685 | 0.7094 | 0.7378 | |
| | 1 | 6 | 25 | | 0.7302 | 0.7863 | 0.7572 | |
| | CatBoost | | | 77.35% | 0.7745 | 0.7735 | 0.7733 | 0.7736 |
| True Label | | 0 | 1 | | | | | |
| | 0 | 9 | 5 | | 0.7909 | 0.7436 | 0.7665 | |
| | 1 | 3 | 10 | | 0.7581 | 0.8034 | 0.7801 | |

accuracy of 76.1%, a recall of 0.76, and a precision of 0.77, indicating a well-balanced ability to detect both depressed and not depressed individuals. In contrast, AdaBoost and Decision Tree underperformed relative to the other models. The Decision Tree classifier, for instance, recorded the lowest recall for detecting depressed individuals (0.70) and an overall accuracy of 69.7%, with 10 false negatives, indicating its limited ability to identify those truly experiencing depression. Similarly, AdaBoost recorded a lower accuracy of 67.1% and struggled with false negatives. XGBoost and LightGBM offered solid mid-range performance, each achieving around 76.0% accuracy with comparable precision and recall values.

**Model evaluation on test data**

Fig 1 and Table 2 illustrate the comparative Receiver Operating Characteristic (ROC) curve analysis and evaluation metrics for seven classification models applied to the task of detecting depression. Among all models, the Random Forest classifier achieved the highest ROC score of 0.91, indicating its strong ability to discriminate between depressed and not depressed individuals. The ROC curve for Random Forest also consistently lies above those of other models, confirming its superior true positive rate across various thresholds.

XGBoost and LightGBM followed closely, each recording an ROC score of 0.86 and 0.83, respectively. Gradient Boosting and CatBoost also demonstrated solid discriminative power, with AUCs of 0.820 and 0.81, respectively. AdaBoost, despite achieving a relatively high precision (0.85), recorded a slightly lower ROC score of 0.81, reflecting a modest trade-off in its overall ability to separate classes. The Decision Tree model, in contrast, recorded the lowest ROC score of 0.66, suggesting weak classification boundaries and a limited ability to distinguish between the two classes.

The ROC analysis shows the results from traditional metrics (accuracy, precision, recall, F1-score), indicating Random Forest as the most robust model for depression detection in this dataset. Ensemble models consistently outperformed

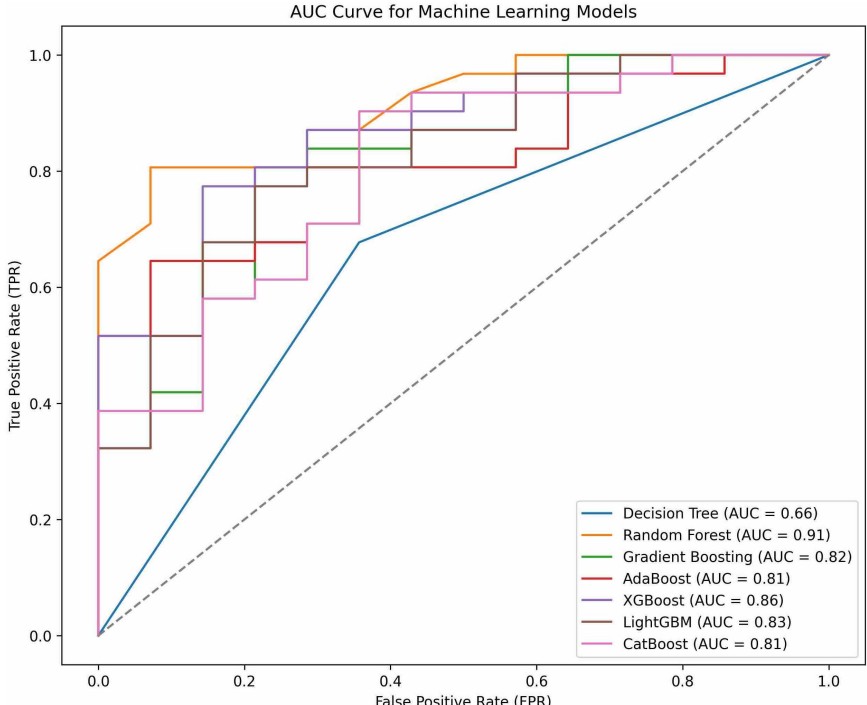

**Fig 1. Receiver Operating Characteristic (ROC) curves for seven machine learning classifiers used to predict depression.** The Random Forest model achieved the highest AUC (0.91), indicating superior discriminatory ability between depressed and not depressed individuals. Ensemble methods (Decision Tree, Random Forest, Gradient Boosting, XGBoost, LightGBM, CatBoost) consistently outperformed the Decision Tree baseline (AUC = 0.66).

**Table 2. Performance metrics for seven classification models in predicting depression were evaluated on the test dataset. Metrics reported include accuracy, precision, recall, F1-score, and the area under the ROC curve (AUC). Random Forest and CatBoost achieved the highest overall accuracy (82.2%), with Random Forest obtaining the highest recall (0.94) and AUC (0.91), indicating its strong ability to identify depressed individuals.**

| Classifier | Accuracy | Precision | Recall | F1 score | ROC AUC Score |
|---|---|---|---|---|---|
| Decision Tree | 0.6667 | 0.8077 | 0.6774 | 0.7368 | 0.6601 |
| Random Forest | 0.8222 | 0.8286 | 0.9355 | 0.8788 | 0.9090 |
| Gradient Boosting | 0.7778 | 0.8621 | 0.8065 | 0.8333 | 0.8203 |
| AdaBoost | 0.7111 | 0.8462 | 0.7097 | 0.7719 | 0.8088 |
| LightGBM | 0.7778 | 0.8621 | 0.8065 | 0.8333 | 0.8272 |
| CatBoost | 0.8222 | 0.8485 | 0.9032 | 0.8750 | 0.8065 |
| XGBoost | 0.8000 | 0.8667 | 0.8387 | 0.8525 | 0.8641 |

the standalone Decision Tree classifier, validating the effectiveness of ensemble learning in improving model stability and predictive power for binary mental health classification tasks.

## Feature importance

Table 3 displays the feature importance analysis across seven machine learning models, reiterating that certain psychological indicators consistently contribute to predicting depression. Model-specific feature importance plots are presented in S2 Fig, with detailed feature descriptions and their corresponding psychological domains outlined in Tab these, ExtSocialIso2, which measures the frequency of in-person interactions with friends or relatives outside the household, emerged as the most robust predictor, appearing in all models and consistently ranked first. PSS14, which reflects the perception that difficulties are overwhelming and unmanageable, was identified by every model and typically ranked among the top three features. Other predictors included StigmaSSB9 (loss of housing due to sexual orientation), BRScale6 (difficulty recovering from setbacks), and PSS12 (persistent task-related rumination), all of which reflect chronic psychosocial stressors or maladaptive coping. Features such as SCS6 and SCS5, reflect the dynamics of social influence and autonomy within

**Table 3. Top predictive features of depression identified across seven tree-based machine learning models, along with them.**

| Feature | Frequency | Decision Tree | Random Forest | Gradient Boosting | AdaBoost | XGBoost | LightGBM | CatBoost |
|---|---|---|---|---|---|---|---|---|
| ExtSocialIso2 | 7 | 1 | 1 | 1 | 1 | 2 | 1 | 1 |
| PSS14 | 7 | 2 | 2 | 2 | 2 | 5 | 8 | 6 |
| StigmaSSB9 | 6 | 3 | NA | 3 | 3 | 8 | 10 | 10 |
| BRScale6 | 4 | NA | 10 | NA | 8 | 9 | NA | 3 |
| PSS12 | 4 | 9 | NA | 4 | 10 | NA | 7 | NA |
| SCS6 | 5 | 8 | 8 | NA | 5 | NA | 5 | 2 |
| PSS3 | 3 | NA | 9 | 5 | | 3 | NA | NA |
| StigmaSSB6 | 3 | NA | 6 | 10 | NA | 1 | NA | NA |
| IntSocialIso5 | 3 | NA | NA | 6 | 4 | | NA | 5 |
| StigmaGNC2 | 4 | NA | NA | NA | 9 | 4 | 9 | 4 |
| SCS5 | 3 | 7 | 5 | NA | NA | 7 | NA | NA |
| StigmaGNC12 | 1 | NA | NA | NA | NA | 2 | NA | NA |
| Middle Income Category | 1 | NA | NA | NA | NA | 6 | NA | NA |
| IntSocialIso13 | 2 | NA | NA | NA | NA | NA | 2 | 8 |

NA: This feature was not ranked in the top 15 features for that model.

one's social network. Stigma-related variables such as StigmaSSB6 (family rejection) and StigmaGNC2 (gender non-conformity bias) were consistently ranked among the top features (S3 Table). Internal perceptions of community belonging and trust, captured by IntSocialIso5 and IntSocialIso13, further contributed to depression classification. The models also identified that individuals in the Income_Category_Middle group, earning between 1,000–4,999 GHS per month, were associated with a higher likelihood of depression.

Using only perceived stress (PSS_StressLevel), external social isolation (ExtSocialIso_Level), and internal social isolation (IntSocialIso_Level), we re-fit two classifiers under the same pipeline (stratified 80/20 split; 5-fold CV on training; one-hot encoding; SMOTE on training folds only). On the held-out test set, the Random Forest (3 predictors) achieved AUC = 0.69 and PR-AUC = 0.72 (F1 = 0.74, recall = 0.87, precision = 0.65, accuracy = 0.66), outperforming the Logistic Regression (3 predictors) baseline (AUC = 0.60, PR-AUC = 0.66; F1 = 0.69, recall = 0.78, precision = 0.62, accuracy = 0.61). Compared with the full-feature Random Forest (AUC ≈ 0.91, F1 ≈ 0.88), the three-predictor model showed an AUC ≈ of −0.22 and F1 ≈ of −0.14, indicating that while stress, social isolation, and belonging, additional features provide complementary information that materially improves discrimination.

## Discussion

The aim of this study was to apply machine learning (ML) techniques to identify key psychosocial predictors of depression among Ghanaian MSM and to evaluate the performance of multiple tree-based classification models. The ML models developed in this study showed comparable performance in predicting depression among MSM in Ghana. Applying seven distinct classification models, Random Forest, Gradient Boosting, AdaBoost, XGBoost, LightGBM, and CatBoost allowed for a comparative assessment of their effectiveness in addressing this important public health issue.

The evaluation of seven machine learning classifiers for depression detection revealed notable differences in their predictive performance. Random Forest consistently emerged as the top-performing model, achieving the highest test accuracy (82.2%), recall (0.94), and ROC AUC score (0.91). These metrics indicate its strong capacity to identify individuals with depression while maintaining a low false-negative rate. Such performance is particularly important in mental health screening, where failure to identify actual cases of depression can delay intervention. Recent studies have similarly demonstrated the robustness of Random Forest in mental health classification tasks, attributing its effectiveness to its ensemble nature and resilience to overfitting [41,42]. The high AUC of 0.91 in the random forest must be interpreted cautiously. While the model incorporated validated psychosocial predictors of depression, such as perceived stress and social isolation, known to be highly correlated with depressive symptoms, this level of performance may not generalise to external populations. The PHQ-2 outcome is closely tied to these constructs, which could partly explain the strong internal signal. Moreover, the relatively small sample size and the complexity of the ensemble methods used may have led to over-fitting despite the use of stratified sampling, SMOTE, and cross-validation.

CatBoost and XGBoost also performed competitively, with CatBoost achieving the same accuracy as Random Forest (82.2%) and a high F1-score (0.87). At the same time, XGBoost recorded the highest precision (0.87) for psychological interpretations and a strong ROC AUC. These results are in line with current literature showing that boosting algorithms excel in tabular health datasets by capturing complex, non-linear relationships and handling feature interactions effectively [43–45]. Their consistent performance across precision, recall, and AUC makes them suitable for clinical decision-support tools in digital mental health platforms. LightGBM and Gradient Boosting offered solid middle-tier performance, with accuracy scores ranging from 77% to 78% and balanced precision-recall trade-offs. Their ROC AUC scores (above 0.82) affirm their ability to distinguish between depressed and not depressed individuals. Although achieving high precision (0.85), AdaBoost underperformed in terms of recall, making it less reliable in scenarios where detecting all true positives is crucial. The Decision Tree model showed the weakest performance across all metrics, with a test accuracy of 66.7% and the lowest AUC score (0.66). This reinforces previous findings that single-tree models lack the complexity to capture nuanced patterns in psychological data [46]. Ensemble models, by contrast, mitigate overfitting and improve generalizability

through combining multiple learners, a well-documented benefit in clinical machine learning research [47–49]. The classification models, particularly Random Forest, CatBoost, and XGBoost, demonstrated superior classification performance in detecting depression. Their ability to maintain high accuracy while minimizing false negatives makes them ideal candidates for real-world deployment in digital health screening applications. These findings support recent trends advocating for ensemble learning as a core strategy in machine learning applications for behavioural and psychological health monitoring.

The results of this study revealed that features related to social isolation, perceived stress, and identity-based stigma were the most consistent and influential predictors of depression among MSM in Ghana. ExtSocialIso2, which measures the frequency of in-person interactions with friends and relatives, emerged as the most important predictor across all models, implying the critical role of social connectedness in protecting against depression. This aligns with broader evidence that social support acts against mental health challenges, particularly when individuals face marginalization or discrimination [22,50]. Similarly, PSS14, which captures the sense of being overwhelmed by difficulties, was consistently ranked highly, underscoring the central role of perceived stress in the mental health of MSM. Stigma-related items such as StigmaSSB9 (loss of housing due to sexual orientation), StigmaSSB6 (family rejection), and StigmaGNC2 (gender non-conformity stigma) were also among the top predictors. It supports the minority stress theory, which posits that sexual minorities experience unique, chronic stressors related to their stigmatized identities that can lead to adverse mental health outcomes [20]. In Ghana, where same-sex behaviour remains criminalised and highly stigmatised, these stressors are intensified and compounded by social, cultural, legal, and structural exclusion [51,52]. The presence of resilience and IntSocialIso5 (a low sense of belonging to one's neighbourhood) among the top features suggests that individual coping mechanisms and perceptions of community support are vital components of psychological resilience. Using only the three top predictors, perceived stress, external social isolation, and internal meaningful predictive signal, but underperformed compared to the full-feature models. This pattern supports the central claim that, despite being the core drivers of depression risk, additional constructs provide complementary information that improves discrimination and overall balance between false positives and false negatives.

From a clinical and public health perspective, the findings show the need for inclusive mental health screening tools that account for experiences of stigma, social isolation, and perceived stress in MSM populations. Interventions should prioritise community-based support systems, reduce social isolation, and address the negative impacts of discrimination. Moreover, integrating stigma-reduction campaigns and sensitisation programs into broader health services may help reduce barriers to care and improve mental health outcomes. In summary, predictive modelling not only strengthens our understanding of the determinants of depression in marginalised populations but also offers a pathway toward more targeted, culturally appropriate mental health interventions in Ghana and similar contexts.

The findings from this study have several important practical implications for mental health support, outreach, and intervention design among MSM in Ghana and other settings. By leveraging machine learning models of exceptionally high-performing classifiers such as Random Forest, CatBoost, and XGBoost, it is possible to develop data-driven, scalable mental health screening tools that are both accurate and sensitive to the unique psychosocial context of MSM communities [53,54].

Given the consistent importance of features related to social isolation, perceived stress, and interpersonal trust, these variables could serve as key components in digital self-assessment tools, such as mobile health (mHealth) apps or online screening platforms. These tools allow MSM to privately and anonymously assess their mental health risk, which is critical in settings like Ghana, where widespread stigma and discrimination discourage in-person help-seeking [52,55,56]. Incorporating these predictors into app-based solutions could enable real-time mental health risk flagging, prompting early intervention and referrals to trusted, affirming community resources. The reduced-feature models tended to favour sensitivity over precision, making them attractive as brief screening tools in settings where missing true cases is costlier than prompting extra follow-up. In practice, the operating threshold can be tuned to local program goals, and a small, judicious

expansion of items, especially those related to stigma and resilience, can recover much of the lost performance while keeping the instruments concise.

MLs' ability to detect risk patterns from simple survey items holds excellent potential for community-based organisations and Non-Governmental Organizations working on sexual minority health-related issues. These groups can use predictive models to prioritise outreach, develop targeted mental health campaigns, and tailor wellness interventions that address the specific needs of MSM, particularly around trust, social support, and emotional coping [8,57].

The insights on feature importance can also be used to train peer educators or community health workers in recognising early indicators of distress, such as social withdrawal or low perceived control, which are not always visible but can be intense precursors of depression. These indicators could be integrated into low-cost, community-based screening protocols, especially in areas with limited access to psychologists or psychiatric care [58–60]. Depression and poor mental health are known to reduce adherence to antiretroviral therapy (ART) and engagement with HIV prevention tools such as Pre-Exposure Prophylaxis (PrEP) [61,62]. Early identification of depression through data-driven models could thus improve overall health outcomes, reduce treatment drop-out, and strengthen health system responsiveness to MSM populations.

## Limitations and future research

While this study presents valuable insights into depression detection using machine learning among MSM in Ghana, some limitations were acknowledged. The dataset size and population scope may limit the generalizability of the findings. Since the study did not adjust hyperparameters, the results indicate baseline feasibility rather than being optimised for a specific model. The sample used may not fully represent the diversity within Ghana's MSM communities in terms of geography, socioeconomic status, or age. While the models utilised structured survey data to capture psychological indicators, the absence of structural and contextual variables such as experiences of discrimination, criminalisation, or access to health services limits the ability to understand the socio-environmental drivers of depression fully. Despite the strong performance of models like Random Forest and XGBoost, interpretability remains a challenge. Although feature importance offers transparency into what variables influence model decisions, black-box models can still limit clinician or community health worker trust and application in real-world screening [49].

## Conclusion

Among the seven classifiers evaluated, Random Forest delivered the best overall performance across accuracy, recall, and ROC-AUC, indicating a strong ability to identify individuals with depression while minimising missed cases. The feature-importance analyses converged on three domains as the most influential and consistent predictors of depression: social isolation, perceived stress, and trust/belonging. These findings align with established psychological theory and underscore the need for culturally sensitive, community-driven screening strategies for MSM in low-resource settings. A reduced-feature sensitivity check using only these three domains retained a meaningful signal but underperformed compared to the full model, suggesting that additional constructs, such as stigma and resilience, add complementary value. We recommend embedding these predictive features in digital or mobile health tools to support earlier detection, tailored guidance, and broader outreach to underserved communities. Looking ahead, work should address interpretability, data diversity (multi-site, longitudinal, and demographically varied samples), external validation and calibration, and co-design with MSM communities to ensure solutions that are accurate, acceptable, and sustainable.

## Supporting information

**S1 Table. Summary of psychological and social constructs included in the model.** Each category represents a group of related variables used to assess key psychosocial domains such as perceived stress, social isolation, behavioral risk, and stigma related to same-sex behavior and gender non-conformity among MSM in Ghana.
(DOCX)

**S1 Fig. Confusion matrices illustrating the performance of six machine learning models.** The models include Logistic Regression, Decision Tree, Random Forest, Naïve Bayes, and K-Nearest Neighbour classifiers, used to predict the severity of depression among MSM in Ghana.
(DOCX)

**S2 Table. Five-fold cross-validation accuracy scores for seven tree-based classification models.** Random Forest achieved the highest mean accuracy (80.36%), followed by CatBoost (77.36%) and XGBoost (76.09%), demonstrating strong generalization performance across folds.
(DOCX)

**S3 Table. Descriptions of the top features identified across tree-based machine learning models.** These features reflect key psychosocial domains including perceived stress (PSS14), external social isolation (ExtSocialIso2), and stigma related to same-sex behavior (StigmaSSB9), among others influencing depression risk.
(DOCX)

**S2 Fig. Top 15 most important features in predicting depression among MSM in Ghana using machine learning classifiers.** External social isolation, perceived stress, and same-sex behavior stigma emerged as the leading predictors across multiple models.
(DOCX)

## Author contributions

**Conceptualization:** Abdulzeid Yen Anafo.

**Formal analysis:** Abdulzeid Yen Anafo.

**Investigation:** Abdulzeid Yen Anafo, LaRon E. Nelson.

**Methodology:** Abdulzeid Yen Anafo.

**Software:** Abdulzeid Yen Anafo.

**Supervision:** LaRon E. Nelson, Leo Wilton, Selasi Ocloo.

**Visualization:** Abdulzeid Yen Anafo.

**Writing – original draft:** Abdulzeid Yen Anafo.

**Writing – review & editing:** LaRon E. Nelson, Leo Wilton, Vincent Uwumboriyhie Gmayinaam, Selasi Ocloo.

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
