## [Decision Letter · Decision Letter 0]

3 Jul 2025

PMEN-D-25-00220

Predicting Depression among Men Who Have Sex with Men in Ghana Using Machine Learning Algorithms

PLOS Mental Health

Dear Dr. Anafo,

Thank you for submitting your manuscript to PLOS Mental Health. After careful consideration, we feel that it has merit but does not fully meet PLOS Mental Health’s publication criteria as it currently stands. Therefore, we invite you to submit a revised version of the manuscript that addresses the points raised during the review process.

Please note that we have only been able to secure a single reviewer to assess your manuscript. We are issuing a decision on your manuscript at this point to prevent further delays in the evaluation of your manuscript. Please be aware that the editor who handles your revised manuscript might find it necessary to invite additional reviewers to assess this work once the revised manuscript is submitted. However, we will aim to proceed on the basis of this single review if possible. 

Please attend to the reviewer's concerns regarding methodological and analytical clarity, as well as clearly stating the rationale & motivation for this study.

We look forward to receiving your revised manuscript.

Kind regards,

Avanti Dey, PhD

Staff Editor

PLOS Mental Health

Journal Requirements:

Additional Editor Comments (if provided):

Reviewers' comments:

Reviewer's Responses to Questions

**Comments to the Author**

1. Does this manuscript meet PLOS Mental Health’s publication criteria?

Reviewer #1: No

2. Has the statistical analysis been performed appropriately and rigorously?

Reviewer #1: No

3. Have the authors made all data underlying the findings in their manuscript fully available (please refer to the Data Availability Statement at the start of the manuscript PDF file)?

Reviewer #1: No

4. Is the manuscript presented in an intelligible fashion and written in standard English?

Reviewer #1: No

Reviewer #1: The presented study applies several classification algorithms to predict depression in a dataset of MSM in Ghana. As a psychologist, I genuinely appreciate research efforts that focus on under-investigated populations and settings like in this paper. The authors deserve credit for addressing an important and often overlooked group. However, as a machine learning (ML) scientist, I must point out several major methodological issues that, in their current form, significantly limit the contribution of this study. Below, I outline my primary concerns:

Major Comments

1. Motive behind the study:

The study’s motivation is weakly developed. The objective of “detecting predictors” aligns more closely with traditional statistical analysis than with a machine learning use case. This is then conflated with “predicting depression” using ML. Why this prediction task is relevant—particularly in the context of interventions, screening, or policy—is barely touched. A clearer justification of the ML angle is needed.

2. Sample size:

The dataset comprises only 200 individuals, which is very limited for a machine learning study. There is increasing evidence that performance estimates based on such small samples carry a high risk of bias and offer limited generalizability. Additionally, most of the included algorithms (e.g., gradient boosting methods) are designed for large, high-dimensional datasets. It remains unclear how such mechanisms are expected to yield performance benefits on only 200 data points. At minimum, a strong justification for the chosen approach despite the sample limitations is required. Also, I believe a simpler baseline algorithm (e.g., penalized logistic regression) should be included for comparison.

Reproducibility and risk of bias:

Especially for small samples, high methodological rigor is essential to avoid inflated and unreliable performance estimates. Unfortunately, the current study lacks transparency and shows signs of potential overfitting. I have the following more specific points

Code availability: Please make the full analysis code publicly available (e.g., via GitHub). This is essential for reproducibility and enables reviewers and readers to assess several unclear methodological aspects.

Data preprocessing:

It appears that imputation and over-/undersampling were conducted prior to the train/test split. If this interpretation is correct, this constitutes data leakage and is methodologically incorrect. Please clarify the order of preprocessing steps.

Train/test split explanation:

The description of the train/test split is very difficult to follow. The manuscript mentions both a 5-fold cross-validation and an 80/20 split—this needs to be clarified and unambiguously described.

Performance variation:

With such a small dataset, performance estimates are likely to vary significantly depending on random initializations or data splits or algorithms. Please report the variation (e.g., standard deviation) across repetitions. Without this, any interpretation of algorithm performance differences remains speculative.

Hyperparameter tuning:

What method was used for hyperparameter tuning? Please describe this clearly.

Evaluation metrics:

Why was accuracy chosen as the main evaluation metric? How was the threshold for binary classification determined? If the default of 0.5 was used, why? The same question applies to other threshold-dependent metrics such as recall.

ROC AUC of 0.91:

The reported AUC of 0.91 raises serious concerns about overfitting. Is there a credible explanation for how depression could be predicted so accurately from sociodemographic variables alone? If the authors believe this result is valid, it should be discussed in depth—particularly in comparison to established findings from high-quality studies using similar data and targets.

Feature importance:

The method used to compute feature importance is not explained. Please clarify.

Minor Comments

Abstract: The abstract is weak. Technical details of the ML pipeline do not belong in the abstract. More essential information is missing, such as the sample size.

Introduction: The introduction lacks a clear structure. I do not understand the relevance of studies on social media-based classification, which are only loosely related to the current approach.

Feature selection: The feature selection method is not clearly explained. Please clarify how features were selected and what rationale guided the process.

**Do you want your identity to be public for this peer review?** For information about this choice, including consent withdrawal, please see our Privacy Policy

Reviewer #1: **Yes: ** Silvan Hornstein

---

## [Decision Letter · Decision Letter 1]

17 Sep 2025

PMEN-D-25-00220R1

Predicting Depression among Men Who Have Sex with Men in Ghana Using Machine Learning Algorithms

PLOS Mental Health

Dear Dr. Anafo,

Thank you for submitting your revised manuscript to PLOS Mental Health. We are pleased to let you know that the original reviewer has approved of the revisions. However unfortunately, as the prior handling editor only secured one reviewer and our policy is to have two reviews prior to acceptance, I have taken over then handling of this paper and secured a second reviewer whose comments you can find below. I am therefore inviting you to revise the manuscript a second time to address the comments of the new reviewer. I am very sorry for this oversight and I understand that it may be very frustrating. Please let me know if you have any concerns. Once you submit the new version, we will get the changes assessed as soon as possible so prevent further delays. 

We look forward to receiving your revised manuscript.

Kind regards,

Dr Karli Montague-Cardoso

Executive Editor

PLOS Mental Health

Journal Requirements:

1. In the online submission form, you indicated that “All relevant data supporting the findings of this study are available upon request from the corresponding author.”.

a) In a public repository, 

b) Within the manuscript itself, or 

c) Uploaded as supplementary information.

2. Please provide separate figure files in .tif or .eps format only and ensure that all files are under our size limit of 10MB.

For more information about how to convert your figure files please see our guidelines: https://journals.plos.org/mentalhealth/s/figures

3. We have noticed that you have uploaded Supporting Information files, but you have not included a list of legends. Please add a full list of legends for your Supporting Information files before or after the references list.

Additional Editor Comments (if provided):

Reviewer #1:

Reviewer #2:

Reviewers' comments:

Reviewer's Responses to Questions

**Comments to the Author**

Reviewer #1: All comments have been addressed

Reviewer #2: (No Response)

publication criteria?

Reviewer #1: Partly

Reviewer #2: Yes

3. Has the statistical analysis been performed appropriately and rigorously?

Reviewer #1: Yes

Reviewer #2: I don't know

4. Have the authors made all data underlying the findings in their manuscript fully available (please refer to the Data Availability Statement at the start of the manuscript PDF file)?

Reviewer #1: No

Reviewer #2: Yes

5. Is the manuscript presented in an intelligible fashion and written in standard English?

Reviewer #1: Yes

Reviewer #2: Yes

Reviewer #1: (No Response)

Reviewer #2: This study focused on applying machine learning models to identify predictors of depression in a sample of 225 men who have sex with men in Ghana.

Comments:

1. Consider: "The full dataset was initially partitioned into training (80%) and testing (20%) sets using stratified sampling to maintain the distribution of the target variable. All data preprocessing steps (imputation, SMOTE, encoding, and scaling) were applied only to the training set to prevent data leakage. The training set was then used to develop the models and tune hyperparameters using stratified 5-fold cross-validation. Model performance was evaluated on the test set, which remained untouched throughout the training process.";

This text is very confusing:

a) It says 80/20 and, after that, 5-fold cross-validation. Is it nested cross-validation? If yes, make it clear.

b) To work, data preprocessing steps (exception for SMOTE) must be 'applied' to the test dataset too. Perhaps the authors would like to say that they were defined/calculated based on the training dataset.

c) It's not clear why you needed to apply SMOTE. What is the original/initial class balance?

2. What is 'error rate'? I think it is the same than accuracy.

3. There is no reason to explain the ML models in different (sub)sections.

4. ROC curve of the decision tree model (figure 1) looks wrong.

5. When analysing results in sections 'Classification report on training data' and 'Model evaluation on test data', they seem not to present any overfitting. Do you agree?

6. There is no reason to state "Future work involving larger and independent datasets will explore model optimization techniques more extensively" in the methods section.

7. Remove "For tree-based models (Decision Tree, Random Forest, Gradient Boosting, AdaBoost, XGBoost, LightGBM, and CatBoost)" - lines 223-224; all models are based on tree.

8. Here is your main conclusion: "The feature importance analysis revealed that social isolation, perceived stress, and trust in others were the most consistent and influential predictors of depression."; have you checked how the models work using only these three variables as predictors?

9. Now, I'll focus on a previous comment from another reviewer and authors' reply.

- Consider "Comment: Hyperparameter tuning-What method was used for hyperparameter tuning? Please describe this clearly.

Response: We thank the reviewer for highlighting the need for clarity regarding hyperparameter tuning.

In the current study, we used default parameters for all classifiers to establish baseline comparisons among tree-based models. This choice was intentional, as our goal was to assess the feasibility of applying various machine learning models for depression risk prediction in a small dataset, rather than to maximize performance for a single model through extensive tuning.";

- This is conflicting, because you provide the following information in the manuscript "The training set was used for model training and hyperparameter tuning, while the test set was reserved for performance evaluation on unseen data."

**Do you want your identity to be public for this peer review?** For information about this choice, including consent withdrawal, please see our Privacy Policy

Reviewer #1: No

Reviewer #2: No

---

## [Decision Letter · Decision Letter 2]

15 Oct 2025

Predicting Depression among Men Who Have Sex with Men in Ghana Using Machine Learning Algorithms

PMEN-D-25-00220R2

Dear Dr Anafo,

We are pleased to inform you that your manuscript 'Predicting Depression among Men Who Have Sex with Men in Ghana Using Machine Learning Algorithms' has been provisionally accepted for publication in PLOS Mental Health.

Best regards,

Karli Montague-Cardoso

Staff Editor

PLOS Mental Health

Reviewer Comments (if any, and for reference):

Reviewer's Responses to Questions

**Comments to the Author**

Reviewer #2: All comments have been addressed

publication criteria?

Reviewer #2: Yes

3. Has the statistical analysis been performed appropriately and rigorously?

Reviewer #2: Yes

4. Have the authors made all data underlying the findings in their manuscript fully available (please refer to the Data Availability Statement at the start of the manuscript PDF file)?

Reviewer #2: Yes

5. Is the manuscript presented in an intelligible fashion and written in standard English?

Reviewer #2: Yes

Reviewer #2: The authors answered my questions and addressed my concerns.

**Do you want your identity to be public for this peer review?** For information about this choice, including consent withdrawal, please see our Privacy Policy

Reviewer #2: No
